# Huntingtin Lowering Strategies

**DOI:** 10.3390/ijms21062146

**Published:** 2020-03-20

**Authors:** Franz Marxreiter, Judith Stemick, Zacharias Kohl

**Affiliations:** 1Huntington’s Disease Outpatient Clinic, Department of Molecular Neurology, University Hospital Erlangen, Schwabachanlage 6, 91054 Erlangen, Germany; judith.stemick@uk-erlangen.de; 2Center for Rare Movement Disorders, Department of Molecular Neurology, University Hospital Erlangen, Schwabachanlage 6, 91054 Erlangen, Germany; 3Department of Neurology, University of Regensburg, Universitätsstraße 84, 93053 Regensburg, Germany; zacharias.kohl@ukr.de

**Keywords:** Huntington’s Disease, huntingtin-lowering, antisense oligonucleotides, RNA interference, disease modification, Chorea, HD

## Abstract

Trials using antisense oligonucleotide technology to lower Huntingtin levels in Huntington’s disease (HD) are currently ongoing. This progress, taking place only 27 years after the identification of the Huntingtin gene (*HTT*) in 1993 reflects the enormous development in genetic engineering in the last decades. It is also the result of passionate basic scientific work and large worldwide registry studies that have advanced the understanding of HD. Increased knowledge of the pathophysiology of this autosomal dominantly inherited CAG-repeat expansion mediated neurodegenerative disease has led to the development of several putative treatment strategies, currently under investigation. These strategies span the whole spectrum of potential targets from genome editing via RNA interference to promoting protein degradation. Yet, recent studies revealed the importance of huntingtin RNA in the pathogenesis of the disease. Therefore, huntingtin-lowering by means of RNA interference appears to be a particular promising strategy. As a matter of fact, these approaches have entered, or are on the verge of entering, the clinical trial period. Here, we provide an overview of huntingtin-lowering approaches via DNA or RNA interference in present clinical trials as well as strategies subject to upcoming therapeutic options. We furthermore discuss putative implications for future treatment of HD patients.

## 1. Introduction

### 1.1. Huntington’s Disease

Huntington’s disease (HD) is clinically characterized by a triad of movement disorders, psychiatric symptoms and cognitive deficits. While the term Huntington’s chorea focuses solely on the hyperkinetic movements, chorea is not the only movement disorder in HD. Dystonia and bradykinesia are common as well [1]. They may even be the sole movement abnormality, especially in juvenile HD. The disease manifests as a movement disorder in about 60% of cases, cognitive deficits dominate the initial clinical picture in 14% and psychiatric abnormalities are the initial presentation in about 20% of cases. Initial cognitive deficits consist of executive dysfunction, concentration and attention deficits, and a disturbed ability to plan. Psychiatric symptoms include irritability, aggressive behavior and disturbed impulse control or apathy [2]. However, the diagnosis of clinically manifest HD is currently exclusively based on the presence of motor symptoms [3]. Current pharmacotherapy in HD is limited to symptomatic treatment of movement disorders and psychiatric symptoms. The only approved drug for the therapy of hyperkinesia and dystonia in HD is tetrabenazine [4]. Despite numerous clinical studies on disease-modifying strategies (see e.g., homepage of the Huntington Study Group), thus far, no therapy exists that is able to slow down, stop or even reverse the course of the disease.

### 1.2. Genetics and Pathophyiology of Huntington’s Disease

HD is a rare autosomal dominantly inherited neurodegenerative disease with worldwide prevalence of 1–10/100,000 [5]. In Europe, a prevalence of 3/100,000 is assumed, the highest prevalence occurs in Sicily with 9.8/100,000 affected islanders [6]. However, current studies show that the genotype frequency with intermediate (36–39) and pathological (>39) expansion in the huntigtin gene (*HTT*) may be as high as 139/100,000 [7]. Genetically, HD belongs to the group of trinucleotide repeat expansion disorders. While individuals with a CAG repeat length of less than 35 located on exon 1 of *HTT* remain asymptomatic, alleles with 40 or more repeats show full penetrance. Intermediate alleles (36–39 CAG repetitions) bear an increased risk to develop HD. There is an inverse correlation between the age of onset, and the length of the CAG repeat expansion, whereby alleles with more than 70 repetitions always cause a manifestation in adolescence. The mean age of onset of this progressive and life limiting disease is 40 years, the average disease duration is 20 years (overview in [8]). Even in early stages, the quality of life and the social environment of those affected is severely impaired. Transcription of pathologically prolonged *HTT* into prolonged huntingtin (htt) pre-mRNA, its processing to mRNA, and translation into a prolonged protein lead to endoplasmatic reticulum stress (reviewed in [9]), misfolding and aggregation (Figure 1*)*, particularly in neurons. Additionally, several toxic short protein species have been described. Some of these species are the result of proteolytic cleavage [10]. The repeat length of the second allele does not influence disease onset or progression [11]. The CAG repeat expansion alone accounts for 50%–70% of the variance in disease onset. Interestingly, the number of uninterrupted CAG repeats, distinct from the length of the polyglutamine segment of the htt protein, accounts for the age of onset of the disease [12]. Several genetic modifiers, most of which represent DNA repair genes, influence the onset of the disease as well [13]. An additional contributor to pathogenesis may be epigenetic modification; in particular, histone deacetylation (HDAC) has repeatedly been shown to negatively influence disease onset. Thus, HDAC inhibition is a promising disease modifying strategy [14]. 

Splice variants and Repeat Associated Non-AUG (RAN) translation can also lead to neurotoxic isoforms of htt [15]. In addition, there is increasing evidence that a pathologically prolonged htt mRNA leads to defective binding of RNA binding proteins, further contributing to pathogenesis (overview in [15]) (Figure 1B). Prolonged htt protein acts primarily through a toxic gain-of-function, interfering with autophagy, vesicle transport, neurotransmitter release and mitochondrial function (Figure 1C). In summary, this results in the degeneration of neurons, predominantly striatal and cortical GABAergic neurons (reviewed in [8]). Furthermore, inflammatory processes also occur, which trigger the progression of the disease (overview in [16]). Thus, from a pathophysiological standpoint, disease-modifying therapies targeting the pathological processing of htt mRNA, or upstream, appear to be the most promising. 

### 1.3. The Scientific Basis for Gene Expression Modification in HD; From the Discovery of the HD Gene to Clinical Trials

In 1983, the Venezuela Project, a field study at Lake Maracaibo in Venezuela, carried out by the World Hereditary Disease Foundation, succeeded in mapping *HTT* on the short arm of chromosome 4 [17]. Following the first discovery of a trinucleotide expansion disorder (fragile X syndrome) in 1991 [18], a CAG repeat expansion in *HTT* was subsequently identified in 1993 as the disease-causing alteration [19], and genetic testing became possible. Already the first murine model of HD in 1996 [20] was able to recapitulate essential aspects of the disease. In 2000, a tetracycline-regulated conditional mouse model in which the expression of mutated *HTT* was suppressed by doxycycline treatment, showed reversibility of brain pathology and behavioral deficits by transgene suppression [21]. Positive effects on RNA interference were later observed in a murine model [22]. Kordasiewicz et al. [23] succeeded in lowering htt in three murine models of HD by using an antisense oligonucleotide (ASO) in 2012, also achieving an improvement of motor symptoms. Additionally, the authors showed that ASO infusion could reduce htt deposits in brains of a primate model. Thus, this study provided decisive preclinical evidence for the concept of “huntingtin holidays” as the basic principle of a disease-modifying strategy for HD patients [24]. Sensitive and standardized immunoassays, which allow the quantification of mutated htt protein in cerebrospinal fluid [25,26], have become available as biomarkers [27].

## 2. Genetic Engineering Approaches for Disease Modification in HD

Disease modification by lowering htt levels via interfering with transcription has been investigated at different levels. Antisense oligonucleotides (ASOs) [23], single-stranded RNA molecules (ssRNAs), small interfering RNAs (siRNAs) [28] and short hairpin RNA (shRNAs) [22] or artificial micro RNA (miRNAs) [29] were used for this purpose. All of these approaches interfere on RNA level and are able to suppress htt levels in vitro and in animal models. With respect to clinical testing however, the utilization of ASOs is the most advanced strategy (Figure 2). By using ASOs, gene expression is only downregulated transiently, and repetitive application is necessary to maintain constant lowering of htt levels. Therefore, these approaches are not called ”gene therapy,” but referred to as “gene expression modification” or “huntingtin-lowering strategies.” In contrast, a “real” gene therapy may be the permanent introduction of an RNA interfering construct into a brain region, e.g., by a viral vector. The currently most advanced gene-therapeutic approaches in neurology rely on adeno-associated viruses (AAVs, [30]). Through their neurotropism as well as their nonpathogenic nature, they ensure safe long-term expression within transfected cells. Several serotypes are being studied for their suitability as a vector (for a review see: [31]). Using AAVs to deliver a micro-RNA expression machinery which produces the therapeutically necessary micro RNA, suppressing htt RNA is the most advanced gene therapy strategy in HD. Since this constitutes a permanent alteration in a transfected cells gene expression system, the term virus-mediated gene therapy is appropriate. In principle, all these approaches may be allele-specific or allele-unspecific.

### 2.1. Gene Expression Modification Using Antisense Oligonucleotides 

ASOs are short, single-stranded nucleic acids, which are complementary to a functional mRNA (or pre-mRNA). They bind to the complementary mRNA strand via Watson-Crick base pairing. Depending on the target sequence of the mRNA, they may prevent protein biosynthesis, block splicing or inhibit binding of RNA binding proteins. The clinical efficacy of ASOs has been demonstrated in two other rare neurological diseases, spinal muscular atrophy [32] and familial amyloid polyneuropathy [33]. Thus, the way seems to be paved for a similar approach in HD.

Roche’s ASO RO7234292 (RG6042), currently tested in a phase III clinical trial, was initially developed by IONIS Pharmaceuticals Inc. (ISIS443139; IONIS-HTTRx). It has been tested in a Phase I/IIa study [34]. This compound is a second-generation, chemically modified synthetic oligomer complementary to a 20-nucleotide portion of htt mRNA. The chemical modification not only prevents rapid degradation, its lipophilic nature also allows the ASO to cross the cellular, as well as the nuclear membrane. In the nucleus, hybridization of RG6042 with htt pre-mRNA and mRNA leads to endogenous RNase H1-mediated degradation, which prevents translation to htt protein [34]. Since it is an allele-unspecific ASO, the generation of both mutant and wild type htt is downregulated. Tabrizi et al. [34] were able to show that with four intrathecal injections of 120 mg HTTRx (later 267
RG6042) via lumbar puncture every 4 weeks, a reduction of mutant htt protein in the cerebrospinal (CSF) fluid of about 40% was achieved. This effect was persistent during the subsequent 2-month follow-up period. Apart from headaches due to lumbar puncture, no serious adverse events were reported. The drug thus met the safety requirements for consecutive multi-center Phase III testing. An open label extension of the Phase I/IIa trial is currently ongoing. Results from this study regarding functional improvement are not available yet. However, it could be shown that a longer injection interval of eight weeks is sufficient to ensure the desired reduction of the mutant htt in cerebrospinal fluid.

The currently ongoing Phase III trial (called Generation HD 1, NCT03761849, Sponsor: Hoffmann-La Roche) investigates, for the first time in HD, an ASO for its potential to slow down the progression of the disease (Figure 1B) in a worldwide study at 101 sites. This randomized, double-blind, placebo-controlled trial will enroll 804 participants. The primary endpoint in the USA is everyday function, measured by the Unified Huntington’s Disease Rating Scale (UHDRS) total function capacity (TFC). Outside the United States, the primary endpoint is the composite UHDRS, a combination score that features domains of everyday function, cognition and motor functions, and is calculated from the UHDRS total motor score (TMS), the UHDRS TFC, the symbol digit modality test and the Stroop word reading [35]. In the three-arm study, participants receive either 120 mg RG 6042 or alternately 120 mg RG 6042 and placebo at eight weekly intervals, or constantly placebo over two years. 

In parallel to Generation HD1, Precision-HD1 (NCT03225833, Sponsor: WAVE Lifesciences) and -HD2 (NCT03225846) follow the approach of reducing htt through an ASO. However, the ASOs WVE-120101 and WVE-120102 are allele-specific, targeting mutant htt mRNA, though not affecting the translation of the healthy allele. This is accomplished by the recognition of two single nucleotide polymorphisms (SNPs), present only on the extended *HTT* gene [36]. The first SNP (RS362037) is present in half of all HD patients worldwide. The second SNP (RS362331) is present in about 40% of those affected. A total of two-thirds of the European and US HD population is likely to carry at least one of these two polymorphisms [36]. This implies that not all gene carriers will benefit from this allele-specific therapy, in case this therapy will be available in the future. In contrast to the afore mentioned non-allele specific ASO, there are no preliminary studies in mice or animals available because the targeted SNPs are not present in the murine genome. Thus, no data are available on safety in animal models. However, the ASOs were tested in vitro and showed successful reduction of mutant htt while leaving wild type htt mRNA transcript and protein intact. The clinical phase I/II is currently ongoing and investigates the safety of WVE-12010, and WVE-120102, as well as the target engagement, i.e., the reduction of the mutant htt protein in the cerebrospinal fluid. In this study, the drug is also administered intrathecally by lumbar puncture.

The first phase of Precision-HD1 and Precision-HD2 started in 2017 and WAVE Lifesciences published preliminary data for the Precision-HD2 study at the end of 2019 in a press release [37]. The company stated that WVE-120102 was able to downregulate mutated htt in cerebrospinal fluid by 12%, which is a rather moderate reduction compared to RG6042. Patients received either placebo or study medication at four different doses (2, 4, 8 or 16 mg). The participants were treated for 5 months. All active treatment arms achieved a reduction of approximately 12% compared to placebo. WAVE also announced that the ASO would show a dose-dependent response, yet WAVE has not published enough information to allow a final evaluation. The safety profile of the ASO seems to be good; side effects were not more frequent in the active compared to placebo arms. A new dosage will be introduced in the Precision-HD2 study (32 mg), and a 32 mg arm will be added to Precision-HD1 so that the final results for both studies are expected by the end of 2020.

### 2.2. Divalent siRNA Mediated Huntingtin-Lowering; the Future at Hand?

In contrast to ASOs, siRNAs exert their effect by binding to mRNA in the cell soma. Degradation of the siRNA/mRNA double strand is then mediated via the RNA-induced silencing (RISC) complex. In the past, siRNA mediated suppression of gene expression was locally restricted due to rapid degradation and low diffusion of the constructs. Recently, sophisticated modification of siRNA using a phosphothioate backbone and a divalent siRNA conjugate enabled whole brain penetrance and a 6-month, dose-dependent suppression of htt in mice and monkeys after a single intraventricular injection [38]. These findings may pave the way for a more sophisticated htt-lowering approach due to a simplified dosing regimen. Yet, these results are currently at the preclinical level and Phase I/II testing is necessary to evaluate the safety and target engagement in humans.

### 2.3. Virally Mediated RNA Interference

In addition to huntingtin-lowering by means of ASOs, virus-mediated gene therapies are upcoming treatment options for HD. Adenovirus-associated suppression of htt mRNA is the most advanced approach, while modalities of mutated htt suppression differ. The viral administration is achieved by a single stereotactic injection into the striatum. 

Currently, a Phase I trial is being conducted by UniQure in the United States (NCT0412049). An AAV serotype 5 is used to deliver a genetic construct that allows RNA interference (AMT-130). Two groups receiving either a high dose or a low dose of AMT-130 will be compared with an imitation surgery arm. Outcomes will be the number and type of adverse events but also the duration of persistence of AMT-130 in the brain, measured by levels of DNA and miRNA expression in the CSF of participants. The genetic construct AMT-130 produces an intrinsic miRNA which is supposed to interact with htt mRNA in a non-allele-specific fashion. The viral approach is supposed to allow a single intervention. The published data show that the htt mRNA degrading effect of AMT-130 is based on a 21–23 nucleotide homology to htt RNA. Previously, both allele-specific and allele non-specific RNA interference with several constructs [39] have been tested. The allele non-specific H–12 construct fused with the pri-miR-451 seemed to be the most promising, leading to an approximately 60% reduction of htt RNA and almost 80% reduction of htt protein in vivo after stereotactic injection into the middle striatum [39] of mice. A recent in vitro follow-up study showed a reduction of 57% htt RNA and 68% htt protein in induce pluripotent stem cell derived neurons of HD patients [40]. The authors also excluded off-target effects, arguing that this is primarily achieved by using the pri-miR-451 backbone, which does not produce a passenger strand [40]. Moreover, no dysregulation of the cellular miRNA apparatus through AAV mediated overexpression of miHTT was observed. Additionally, the dysregulation of other genes was negligible and seemed to be due to viral transduction, rather than miRNA expression [40]. 

A further attempt to deliver miRNA-based hairpins using an AAV serotype 2 vector was successful for reduction of mutant htt protein in YAC128 mice [41]; this approach is currently further developed by Voyager Therapeutics.

### 2.4. Virally Mediated Suppression of Mutant Huntingtin Transcription Using Zink-Finger Transcription Factors

Recently, a different virus-based gene therapeutic approach was published [42]. An AAV was used to deliver Zinc finger protein transcription factors linked to the krüppel associated box KRAB transcriptional repression domain of the human *KOx1* gene. This served as a suppressive DNA binding factor for mutant htt mRNA. The authors could convincingly show allele-specific knock down of mutant htt protein in a dose-dependent fashion by some of these synthesized vectors. At higher doses, a complete suppression of mutated htt protein was achieved in vitro. In vivo, a 50% reduction of mutant htt protein was shown in three murine models with almost unchanged expression of the wild-type allele. Especially in a murine model with only 50 CAG-repeat expansions, an expansion in the clinically relevant range, significant htt reduction was achieved. In the well-described progressive R6/2 animal model, the application of an AAV and suppression of mutated htt resulted in a functional improvement, dependent on the timepoint of injection. Even complete normalization of some phenotypes was observed after 12 weeks. The virally applied transgene was active for at least 100 days in culture and at least 9 months in the mouse model. Additionally, the authors showed a 99% knockdown in fibroblasts and induced pluripotent stem cell derived primary neuron cultures of HD patients with a wide dose range. The wild-type allele was unaffected by this knockdown. Other CAG repeat expansions were only slightly altered by this approach. Thus, a new therapeutic concept for htt expression regulation, interfering with the transcription of mutant *HTT*, was established in this study. To what extent clinical studies will follow is not known at this time.

## 3. Discussion

### 3.1. The Age of Gene Expression Modification Has Begun. Does It Begin for HD?

Generation HD1, and Precision-HD1 and -HD2, represent some of the most exciting clinical studies in neurology at present, and further promising approaches for disease modification in HD are to be expected (for an in depth review, see also [43]). ASO-based therapies have already been approved for several diseases including both aforementioned neuromuscular diseases. Moreover, a variety of further ASOs are currently being investigated in clinical trials for other neurological disorders [44]. The specific interest in the Generation HD1 trial is based on the question whether it will be possible to achieve a modification of a progressive neurodegenerative disease by lumbar intrathecal application of an ASO. Assuming a positive outcome, the important question arises which functional consequence a permanent reduction of htt will have.

### 3.2. Is Huntingtin Expression Modification Safe?

Huntingtin is an indispensable protein in embryonic brain development. Additionally, it appears to interact with a huge amount of proteins [45], suggesting multiple roles as a scaffold protein. Yet, its precise physiological role in the adult brain remains less well understood. Lowering wild type htt levels might interfere with the many cellular functions described, such as axonal transport, transcriptional regulation and neuronal survival. Several murine studies addressing the effect of *HTT* knockdown have been performed, delivering conflicting results, depending on the time of knockdown and tissue-specificity [46,47]. However, adult knockdown in the central nervous system CNS appears to be well-tolerated [47]. In summary, the question of negative effects of permanently lowered htt levels in the adult cannot be answered, yet [48]. Thus, many important theoretical risks exist, also in the adult brain. Nevertheless, the findings in carriers of homozygous CAG repeat expansions who develop normally and show similar age of onset as heterozygous CAG repeat expansion carriers suggests that in HD, the loss of wild type htt function may not be relevant [11]. Studies in non-human primates have shown that a reduction of up to 45% htt has no phenotypic effects, and this reduced wild type htt expression is tolerated in the non-human primate striatum [29,49]. Yet, it remains completely unclear if an effective non-allele specific ASO-based therapy might show detrimental effects after 10 or more years. It is important to mention that currently, no test to determine the amount of wild type htt in CSF is available yet, due to the fact that available assays to detect mutant huntingtin rely on immunoassays that first precipitate total huntingtin followed by a detection steps recognizing the polyglutamine stretch of the protein [25,26]. Moreover, what remains unresolved is the question whether ASO treatment needs to be applied life long to achieve long-term stability of clinical effects, or if injection intervals can be extended or probably even longer treatment pauses are possible. This aspect touches not only safety issues regarding repetitive CSF injections, but is also important with respect to issues concerning long-term costs (see below). While an allele-unspecific approach may cause ethical problems as a treatment option in premanifest HD, an allele-specific approach may also enable addressing the pre-manifest or asymptomatic HD population in the future. Here, in case of premanifest gene carriers, an important question is the amount of mutant htt lowering, necessary to slow development/progression of disease. The 12% reduction that WAVE has reported, compared to the 40%–60% reduction of RG6042, makes a difference of about 40%. How this affects disease activity is completely unknown. In summary, we currently do not know whether a specific reduction of mutant htt is inferior or superior to a general suppression of htt. This is nonetheless information that will only be available after both approaches have been studied. Furthermore, it should be mentioned that absolute doses of RG6042 with 120 mg are fundamentally different from the doses in Precision-HD 2 (16 mg). Possibly, higher doses in Precision-HD 2 could also achieve a higher rate of htt reduction.

### 3.3. Putative Problems of the Allele-Specific Approach

The currently tested allele-specific approaches rely on SNPs that are prevalent particularly in Northern American and European populations [36]. A detailed characterization of *HTT* haplotypes in other populations has only recently been performed [50]. It suggests that the current allele-specific approaches fail to target Southern European, Southern Asian and Middle Eastern populations. Different haplotypes appear to be needed to enable high treatment rates for people with HD worldwide [50]. Thus, population-specific ASOs are most likely necessary to enable a worldwide availability of an allele-specific ASO approach. Formulated in a pointed way, a single allele-specific approach intrinsically leads to the benefit of a selected population, discriminating against others, if no strategy is pursued that ensures the development of allele-specific approaches with the intent of covering a maximal population worldwide. Here, socio-economic aspects may of course play a major role. In this regard, it is noteworthy that recently, an individualized ASO therapy has been realized within a year [51]. Thus, from a biotechnological standpoint, the generation of populations-specific ASOs is feasible. Yet, assuming that a few ASO molecules are needed to equally treat HD worldwide, current regulations in most countries would make sequential clinical testing necessary for each molecule. In addition, genotyping to determine the appropriate treatment will become necessary [52]. 

### 3.4. Can We Deal with the Expenses for a New Therapy?

Of high practical importance is the application frequency of the ASO or other huntingtin-lowering therapies. Intrathecal applications, performed on a bimonthly basis, mean a significant clinical and medical-economic expenditure. With regard to public health costs, it is worth taking a look at the two approved ASO therapies in neurology. For Spinraza, the treatment of spinal muscular atrophy (SMA), the costs amount to approximately 620,000 € in the first year of treatment and 310,000 € in the second, in Germany [53]. Patisiran’s therapy for familial amyloidic polyneuropathy (FAP) costs between 344,000 €, and 515,495 € per year [54]. It should be anticipated whether it is ethically justifiable to withhold such a therapy from mutation carriers, even if it has shown effects only in clinically-manifested patients. This coincides with the question at which time point it makes sense pathophysiologically to start a gene expression modification-based therapy. Since neurodegeneration precedes motor impairment [8], is it tolerable to withhold a therapy from a mutation carrier before obvious motor signs (on which the diagnosis of manifest HD is currently based) appear? How can we reliably record this prodromal phase of the disease with partly intermittent and elusive motor, psychiatric and cognitive symptoms? Methods exist which, based on CAG repetition length and age, offer possibilities to predict the period of conversion from premanifest to manifest HD [55,56]. Yet, these concepts have been established to capture risk populations for studies. Thus, CAG repeat expansion-based risk stratification does not allow a precise individual prognosis due to the considerable variance in disease onset [13]. Recently, another prognostic index for HD has been published, based on data from REGISTRY, Track-HD and the Cooperative Huntington Observational Trial. This score gives reasonable probabilities for conversion from premanifest to manifest HD, but requires the symbol-digit-modality test and a full UHDRS motor score [57]. Here, digital biomarkers may represent another modality for early detection of prodromal HD. If any huntingtin-lowering strategy will be approved, long-term monitoring of the effects of reduced htt levels is of tremendous interest and should be realized by follow up registers of patients receiving the therapy. Here, collaborative efforts of pharmaceutical companies providing these therapies and the HD research community would be an important basis. 

### 3.5. If Studies Fail

Assuming negative study results, the data obtained will nevertheless be of high relevance for the scientific community. The knowledge of side effects, as well as the analysis of the temporal course of htt levels in the CSF and their correlation to imaging findings represents immensely valuable information, in particular for future therapeutic strategies. The implementation of digital biomarkers as explorative secondary endpoints in Generation HD1 will also provide important data, since little is known about the long-term acceptance of such concepts, nor about the actual significance in terms of “sensitivity-to-change” of such methods.

## Figures and Tables

**Figure 1 ijms-21-02146-f001:**
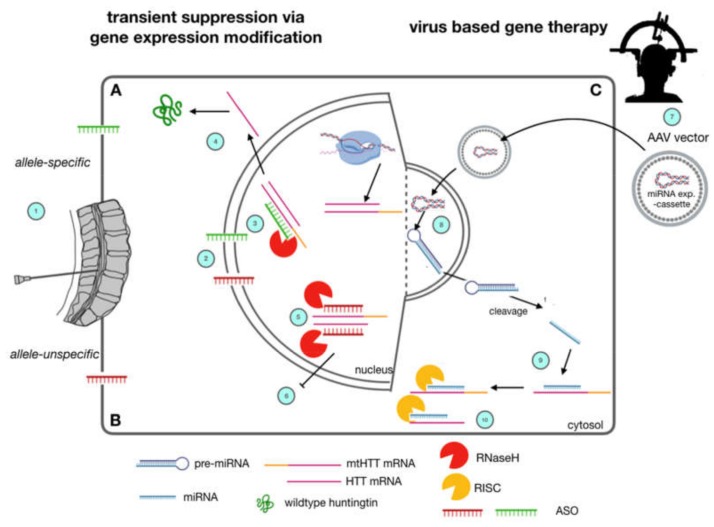
Huntingtin-lowering strategies currently under clinical investigation in Huntington’s Disease. Allele-specific (**A**) and allele-unspecific (**B**) Antisense Oligonucleotide (ASO) strategies as well as virally mediated strategies (**C**) are currently being evaluated in clinical trials. ASOs (A, B) are applied via lumbar puncture (1). They are supposed to reach the cerebral cortex and deeper brain regions via diffusion along the neuraxis through cerebrospinal fluid CSF turnover. They enter brain cells and nuclei due to their lipophilic backbone modification (2). By Watson-Crick base pairing, they bind to both mutant htt and wild type htt (5, allele-unspecific) or mutant htt (3, allele-specific). This leads to nuclear RNaseH mediated degradation of pre-mRNA and mRNA and a subsequent lowering in htt protein levels of wt and mt htt (6) or mt htt, only (4). Gene-Therapeutic strategies (C) rely on a single stereotactic injection into the striatum (7) of adeno-associated viruses carrying a mi-RNA expression cassette, which is able to enter neuronal nuclei, and express a miRNA construct (8). After cleavage, this construct suppresses expression of mutant htt by RNA interference (9) and subsequent degradation by the cytosolic RISC complex (10).

**Figure 2 ijms-21-02146-f002:**
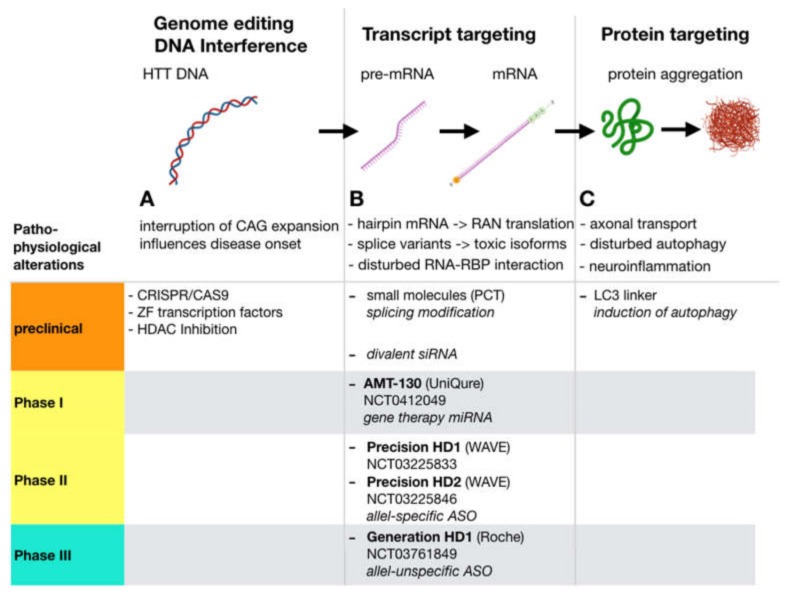
Huntingtin- lowering approaches: Along the process from htt DNA transcription via RNA processing and translation to protein folding and function, several well-defined points of interference are possible that ultimately result in lower levels of total or mutated htt protein. Strategies that aim at the protein level (C), i.e., by means of induction of autophagy rely on the hypothesis that HD is mainly a proteinopathy that results from disturbed function and neurotoxic accumulation of mutant htt protein. In contrast, gene expression modification strategies (B) inhibit the generation of the protein, but also may be suitable to cover pathologic aspects on the RNA level that may significantly contribute to HD pathogenesis. (A) Genome editing via, i.e., CRISPR/Cas9 may permanently correct mutant *HTT*. Correction of the CAG Repeat expansion may have additional beneficial effects since the length of the uninterrupted CAG repeat length on DNA level is inversely correlated to disease onset (for references see text). Altering metagenomic structure by HDAC inhibition, or using Zink Finger transcription factors can inhibit transcription of mutant DNA to mRNA. PCT: PTC Therapeutics; LC3: autophagosome protein microtubule-associated protein 1A/1B light chain 3.

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
