# Peer review of "Huntingtin Lowering Strategies"

_ijms, 2020, doi:10.3390/ijms21062146_

Round 1

Reviewer 1 Report

This is a great review about the therapeutic strategies in development to lower expanded htt in HD. I believe it will be a great addition to the literature. 

As a very minor point, in the introduction about the causes of the diseases, the authors could mention endoplasmic reticulum stress and epigenetic modifications as driver of the disease onset. There is extensive literature about both. 

Beside this, I really enjoyed the manuscript. Figures are also very nice. 

Author Response

We'd  like to thank the reviewer very much for the positive feedback. We are delighted to add the two aspects mentioned to our introduction and now write in lines 64-67: 

Transcription of pathologically prolonged HTT into prolonged huntingtin (htt) pre-mRNA, its processing to mRNA, and its translation into a prolonged protein leads to endoplasmatic reticulum stress (reviewed in [9]) misfolding and aggregation (figure 1), particularly in neurons.

and in lines 73-76: 

An additional contributor to pathogenesis may be epigenetic modification, in particular histone deacetylation has repeatedly been shown to negatively influence disease onset, and HDAC Inhibition is a promising disease modifying strategy [14].  

We cited two recent reviews on the topic that allow the reader to get informed on both aspects, which in itself are topics worth writing a review:

9:Jiang, Y.; Chadwick, S. R.; Lajoie, P. Endoplasmic reticulum stress: The cause and solution to Huntington's disease? Brain Res 2016, 1648, 650–657.

14: Francelle, L.; Lotz, C.; Outeiro, T.; Brouillet, E.; Merienne, K. Contribution of neuroepigenetics to Huntington’s disease. Front Hum Neurosci 2017, 11, 17.

We hope that all issues could be addressed by our adaptions.

Franz Marxreiter

Reviewer 2 Report

Marxreiter et al review huntingtin-lowering strategies, mainly focusing on those in clinical trials, targeting RNA. The figures are very informative.

My only main concern is what if anything does this review add to the recent comprehensive review “Huntingtin lowering strategies for disease modification in Huntingtons’ Disease” by Tabrizi et al, Neuron 2019 https://www.ncbi.nlm.nih.gov/pubmed/30844400, which is not cited in the current manuscript.

Minor:

1) Line 47: I guess HDStudygroup should be huntingtonstudygroup

2) Line 53: use the thousands separator consistently

3) Line 109: “In this regard the utilization of ASOs is the most advanced strategy” –Unclear in which regard?

4) Line 230-232: Allele-specific or not?

5) Line 235: “passage” should be “passenger”

6) Lines 304-305: please rephrase “if application timepoints can be enlarged”

7) Lines 310-311: please explain why “no test to determine the amount of wild type htt in any biosample, e.g. CSF, is available yet”?

8) Line 328: Spell out SMA

9) Line 329: What is FTP?

10) Discussion: Allele-specific approach should have the advantage of sparing the wild-type copy of htt to do whatever it is supposed to do. Is there any theoretical reason why allele-unspecific approach would have an advantage over allele-specific approach?

11) Fig 1: allele specific and allele-unspecific

12) Fig 2: explain what are PCT and LC3. Can probably spell out “ind”., as there is enough space.

Author Response

My only main concern is what if anything does this review add to the recent comprehensive review “Huntingtin lowering strategies for disease modification in Huntingtons’ Disease” by Tabrizi et al, Neuron 2019https://www.ncbi.nlm.nih.gov/pubmed/30844400, which is not cited in the current manuscript.

A: We thank the reviewer very much for the critical reading of our manuscript and for pointing out several mistakes. We hopefully addressed them in an appropriate fashion.
With respect to novelty, of course reviews on the topic have been written previously in a comprehensive fashion. The mentioned review by Tabrizi et al. is an in depth and excellent summary on the topic. Our intent was to provide the readers of the special issue "CNS Drug Action in Neurodegenerative Diseases" of the International Journal of Molecular Sciences information on huntingtin lowering strategies in ongoing and upcoming clinical trials, since the readers may not be familiar with the advances made in the field. Moreover, our aim was to focus on aspects necessary to gain an informative and solid insight into the field, providing the necessary molecular mechanisms, ongoing clinical trials, and developments in a condensed fashion. We also aimed to include novel bioengineering results into this manuscript (lines 205-214) addressing divalent siRNA htt lowering. Also, we have taken into consideration novel preliminary results from the Precision HD2 study. Finally, as clinicians, we focused parts of our discussion on medico-economical and ethical aspects (chapter 3.3.), an issue often underrepresented. We therefore believe that our review may be an interesting contribution. However, we thank the reviewer very much for pointing to the missing relevant reference. Therefore, we now write (lines 281-283):
Generation-HD1, and Precision-HD1 and -HD2 represent some of the most exciting clinical studies in neurology at present, and further promising approaches for disease modification in HD are to be expected (for an in depth review see also [43].

43: Tabrizi, S. J.; Ghosh, R.; Leavitt, B. R. Huntingtin Lowering Strategies for Disease Modification in Huntington’s Disease. Neuron 2019, 101, 801–819.

Minor:

1) Line 47: I guess HDStudygroup should be huntingtonstudygroup
A: we now write huntingtonstudygroup

2) Line 53: use the thousands separator consistently
A: we now consistently write 100.000

3) Line 109: “In this regard the utilization of ASOs is the most advanced strategy” –Unclear in which regard?
A: We thank the reviewer very much, and now state (lines 112-113): With respect to clinical testing, however, the utilization of ASOs is the most advanced strategy.

4) Line 230-232: Allele-specific or not?
A: The ongoing clinical trial uses a non-specific approach. Yet, the publication by Miniarikova et al. (https://doi.org/10.1038/mtna.2016.7) has investigated both approaches. To state more clearly that a non-specific approach derived from this paper is now used for the Phase 1/2 clinical trial, we now write:
The published data show that the htt mRNA degrading effect of AMT–130 is based on a 21–23 nucleotide homology to htt RNA. Previously, both, allele-specific and allele non-specific RNA interference with several constructs [39] have been shown. The allele non-specific H–12 construct fused with the pri-miR–451 seemed to be the most promising, leading to an approximately 60% reduction of htt RNA and almost 80% reduction of htt protein in vivo after stereotactic injection into the middle striatum [39] of mice.

5) Line 235: “passage” should be “passenger”
A: we now write passenger

6) Lines 304-305: please rephrase “if application timepoints can be enlarged”
A: we now write: if injection intervals can be extended

7) Lines 310-311: please explain why “no test to determine the amount of wild type htt in any biosample, e.g. CSF, is available yet”?
A: we now write (lines 307-310): It is important to mention, that currently no test to determine the amount of wild type htt in CSF, is available yet. Current assays used to detect mutant huntingtin rely on immunoassays that first precipitate total huntingtin followed by a detection steps recognizing the polyglutamine stretch of the protein [25,26].

We hope that this explanation is sufficient. Yet, if the reviewer wishes further clarification, we would be happy to expand this aspect further.

8) Line 348: Spell out SMA
A: we now write: spinal muscular atrophy (SMA)

9) Line 349: What is FTP?
A: We now write: Patisiran’s therapy for familial amyloid polyneuropathy (FAP)

10) Discussion: Allele-specific approach should have the advantage of sparing the wild-type copy of htt to do whatever it is supposed to do. Is there any theoretical reason why allele-unspecific approach would have an advantage over allele-specific approach?

A: This is a very interesting question. Indeed, we unintendedly discussed allele-specific vs allele-unspecific ASO strategies with a tendency towards an inherent advantage of allele-specific approaches. However, this is an assumption that needs to be proven first. We did cover the fact that an allele-specific approach only enables treating a certain amount of the HD population and wrote: "For the allele-specific ASO approaches, the determination of suitable SNPs to treat a significant amount of the HD patient population, with still patients left untreated due to absence of targeted SNPs may be a relevant issue". One may argue, that, an allele-unspecific approach would be available to the whole HD community, while allele-specific approaches may only be used in 2/3rds of HD patients (which in itself is a number that is debatable; see below). We believe that this aspect deserves further discussion. We deleted the previous sentence, and added an additional paragraph:
3.3. Putative problems of the allele-specific approach;
and now write (lines 324-340):
The currently tested allele-specific approaches rely on SNPs that are prevalent particularly in Northern American and European populations [36]. A detailed characterization of HTT haplotypes in other populations has only recently been performed [50]. It suggests that the current allele-specific approaches fail to target Southern European, Southern Asian and Middle Eastern populations. Different haplotypes appear to be needed to enable high treatment rates for people with HD worldwide [50]. Population-specific ASOs are most likely needed to enable a worldwide availability of an allele-specific ASO approach. Formulated in a pointed way, a single allele-specific approach intrinsically leads to a benefit of a selected population, discriminating against others, if no strategy is pursued that ensures the development of allele-specific approaches with the intent of covering the maximal population worldwide. Here, socio-economic aspects may play a major role. In this regard, it is noteworthy that recently an individualized ASO therapy has been realized within a year [51]. Thus, from a biotechnological standpoint, the generation of populations-specific ASOs is feasible. Yet, assuming that a few ASO molecules are needed to equally treat HD worldwide in all populations, current regulations in most countries would make sequential clinical testing necessary for each molecule. In addition, genotyping to determine the appropriate treatment will become necessary [52].     

11) Fig 1: allele-specific and allele-unspecific
A: Thank you very much. We now write allele-specific and allele-unspecific

12) Fig 2: explain what are PCT and LC3. Can probably spell out “ind”., as there is enough space.
A: we now spell out induction, and explain both abbreviations in the figure legend: PCT: PTC Therapeutics; LC3: autophagosome protein microtubule-associated protein 1A/1B light chain 3